# Hox dosage contributes to flight appendage morphology in *Drosophila*

Rachel Paul [1,2], Guillaume Giraud[1], Katrin Domsch [3], Marilyne Duffraisse[1], Frédéric Marmigère[1], Soumen Khan [4], Solene Vanderperre [1], Ingrid Lohmann [3], Robby Stoks [5], L. S. Shashidhara [4,6] & Samir Merabet [1✉]

Flying insects have invaded all the aerial space on Earth and this astonishing radiation could not have been possible without a remarkable morphological diversification of their flight appendages. Here, we show that characteristic spatial expression profiles and levels of the Hox genes *Antennapedia* (*Antp*) and *Ultrabithorax* (*Ubx*) underlie the formation of two different flight organs in the fruit fly *Drosophila melanogaster*. We further demonstrate that flight appendage morphology is dependent on specific Hox doses. Interestingly, we find that wing morphology from evolutionary distant four-winged insect species is also associated with a differential expression of *Antp* and *Ubx*. We propose that variation in the spatial expression profile and dosage of Hox proteins is a major determinant of flight appendage diversification in *Drosophila* and possibly in other insect species during evolution.

[1] IGFL, CNRS UMR5242, ENS Lyon, Lyon, France. [2] Laboratory of Genetics and Development, Institut de Recherches Cliniques de Montréal, Montréal, QC, Canada. [3] University of Heidelberg, Centre for Organismal Studies (COS) Heidelberg Department of Developmental Biology, Heidelberg, Germany. [4] Indian Institute of Science Education and Research (IISER), Pashan Pune, India. [5] Laboratory of Aquatic Ecology, Evolution and Conservation, Leuven, Belgium. [6] Ashoka University, Sonipat, India. ✉email: samir.merabet@ens-lyon.fr

nsects have developed various strategies for flying, which is reflected at the level of their flight appendages. For example, insects can use two pairs of wings with similar or dissimilar morphology (as observed in different orders, including Odonata, Hymenoptera, and Lepidoptera), or only one pair of wings and the other pair of wings transformed as a protective envelope (the elytra in Coleoptera) or a balancing organ (the haltere in Diptera). What are the molecular mechanisms underlying such morphological diversification?

Most of our current understanding stems from work in the fruit fly *Drosophila melanogaster* and the beetle *Tribolium castaneum*. A similar scenario is observed in those two species: the Hox gene *Ultrabithorax (Ubx)* represses the formation of the second thoracic (T2) flight appendage and promotes the development of the third thoracic (T3) flight appendage, while the Hox gene *Antennapedia (Antp)* has no obvious function in flight appendage formation on the T2 segment[1–6]. These observations led to the assumption that Ubx shares a role in T3 flight appendage development, while T2 flight appendage formation is Hox-free in insects[5,7]. Analyses in different butterfly species confirmed the role of Ubx[7,8], but also revealed that Antp could be involved in the evolution of butterfly-specific wing patterns[9]. Along the same line, loss of Ubx was shown to affect the morphology of the forewing (FW, on the T2 segment) and hindwing (HW, on the T3 segment) in planthoppers[10], underlining that both T2 and T3 flight appendages could require the input of Hox genes in some instances.

Here, we show that the wing formation is not a Hox-free state and that different doses of *Antp* or *Ubx* instruct different flight appendage morphologies in the fruit fly *D. melanogaster*. In addition, we observe that insect species with similar or dissimilar pairs of wings express similar or distinct Hox expression levels in their forewing and hindwing primordia, respectively. Our results highlight that Hox dosage is a major determinant of flight appendage morphology in *Drosophila*, and we propose that variation in the spatial expression profile and dosage of Hox proteins could apply more generally for controlling flight appendage diversification during insect evolution.

## Results

### Antp is expressed in the wing primordium and controls wing formation in *Drosophila*.
Pioneering work showed that Antp was neither expressed in the wing primordium region that will give rise to the distal wing blade (the so-called "pouch") nor required for wing blade formation in *Drosophila*[5]. These results were acquired more than two decades ago, with no information on Antp expression in the wing imaginal disc (the wing primordium) of early L1 and L2 larval stages. We thus repeated immunofluorescence assay in the wing imaginal discs, using the original 4C3 or a more recent 8C11 anti-Antp antibody. Both antibodies revealed the previously characterized expression profile of Antp in the hinge and notum regions of the late L3 wing disc (Fig. 1a, b). The 8C11 antibody also revealed a clear additional pattern in the pouch region, in particular in the ventral (V) and dorsal (D) sides (Fig. 1b). Staining at earlier stages showed that Antp was dynamically expressed in the pouch region, with a homogenous distribution at the L1 larval stage that becomes progressively restricted to the V and D sides, and a specific expression in a few cells along the DV boundary of the pouch from L2 to mid-L3 larval stages (Fig. 1c, d). Both the increased sensitivity of the Antp antibody and the improved staining protocol (see "Methods") most likely explain why this expression profile has been missed before. This expression profile of Antp in the wing disc pouch was also reproduced with a GFP antibody recognizing an endogenous Antp-GFP fusion protein that we generated (Fig. 1e, see

"Methods") and with the *P1* autoregulatory element of *Antp*[5,11] (Fig. 1f). In addition, recent single-cell RNA-seq data performed in the wing disc[12] confirmed that *Antp* was co-expressed with several wing-specifying genes in the pouch of L3 wing disc (Supplementary Fig. 1).

Considering the expression profile of Antp in the wing disc pouch, we asked whether it was of any functional relevance for the proper development of the wing. Subtle wing phenotypes have previously been described in mutant clones for *Antp*[13,14], and *Antp* was shown to induce wing-like structures in the head under particular ectopic expression conditions in *Drosophila*[15,16]. Here, we specifically targeted *Antp* expression in the wing disc pouch region by using CRISPR/Cas9 (ref. [17]) with different Gal4 drivers. The specificity of gRNAs against *Antp* was validated in the embryo and haltere and leg imaginal discs (Supplementary Fig. 2). Abolishing *Antp* expression in the whole pouch of the wing disc with the *MS1096-Gal4* driver (Fig. 2a-a′) led to the strong wing size reduction and weak margin defects (Fig. 2c-c′). A similar phenotype was observed when *Antp* was targeted by using RNA interference[18] (Supplementary Fig. 3). Because the wing margin originates from cells located along the DV boundary of the pouch, we specifically targeted *Antp* expression in these corresponding cells by using the *Distalless (Dll)-Gal4* driver (Fig. 2b-b′). Under this condition, we observed strong wing margin defects, but no obvious wing size reduction (Fig. 2c-c′). Finally, depleting *Antp* at different larval stages showed that *Antp* was required for correct wing size and wing margin formation at the L1 and L2 stages, and only for correct wing size at the L3 stage (Fig. 2d-d″). The wing phenotypes induced by the loss of *Antp* upon CRISPR/Cas9 were also correlated with the altered expression profiles of genes involved in wing disc growth (*Mad*, *nubbin (nub)*, and *spalt (sal)*) and wing margin formation (*wingless (wg)*, Fig. 3a-a′). Together these results established that the expression of Antp in the wing disc pouch is required for the proper wing blade formation in the adult fly.

### Homothorax (Hth) is a positive regulator of *Antp* in the wing disc.
To identify potential upstream regulators of *Antp* in the wing disc pouch, we searched for genes displaying similar expression profiles and having a characterized role in wing formation. We found the expression of the Hox cofactor Homothorax (Hth)[19] and Antp to overlap, both in the hinge region and in the D and V parts of the pouch (Fig. 4a). Moreover, Antp expression in the D and V part was lost, when *hth* function was compromised in the wing disc pouch (Fig. 4a), resulting in malformed adult wings (Fig. 4b), while the loss of *Antp* did not affect the expression of Hth (Fig. 4a). In addition, ectopic expression of *hth* in the whole pouch of the wing disc was sufficient to induce high Antp expression levels in the same cells (Fig. 4a). This effect was followed by wing-to-haltere transformation phenotypic traits (Fig. 4b). Finally, the analysis of Hth Chip-seq data performed in the wing imaginal discs[20,21] revealed a significant binding on the *P1 cis*-regulatory region and another putative enhancer of *Antp* (Fig. 4c). Thus, Hth could directly regulate the expression of *Antp* in the wing disc pouch. Interestingly, Hth was recently described as a transcriptional repressor of *Ubx* in the haltere imaginal disc[22], highlighting that this regulator differentially controls Hox gene expression in the different flight primordia.

### Hox dosage controls wing and haltere development in *Drosophila*.
The wing-to-haltere transformation phenotype observed upon the *hth*-induced expression of Antp suggested that the functional output of Antp might depend on its expression profile. The same assumption might apply for Ubx, which is strongly expressed in the entire pouch of the haltere disc (Fig. 5a). By

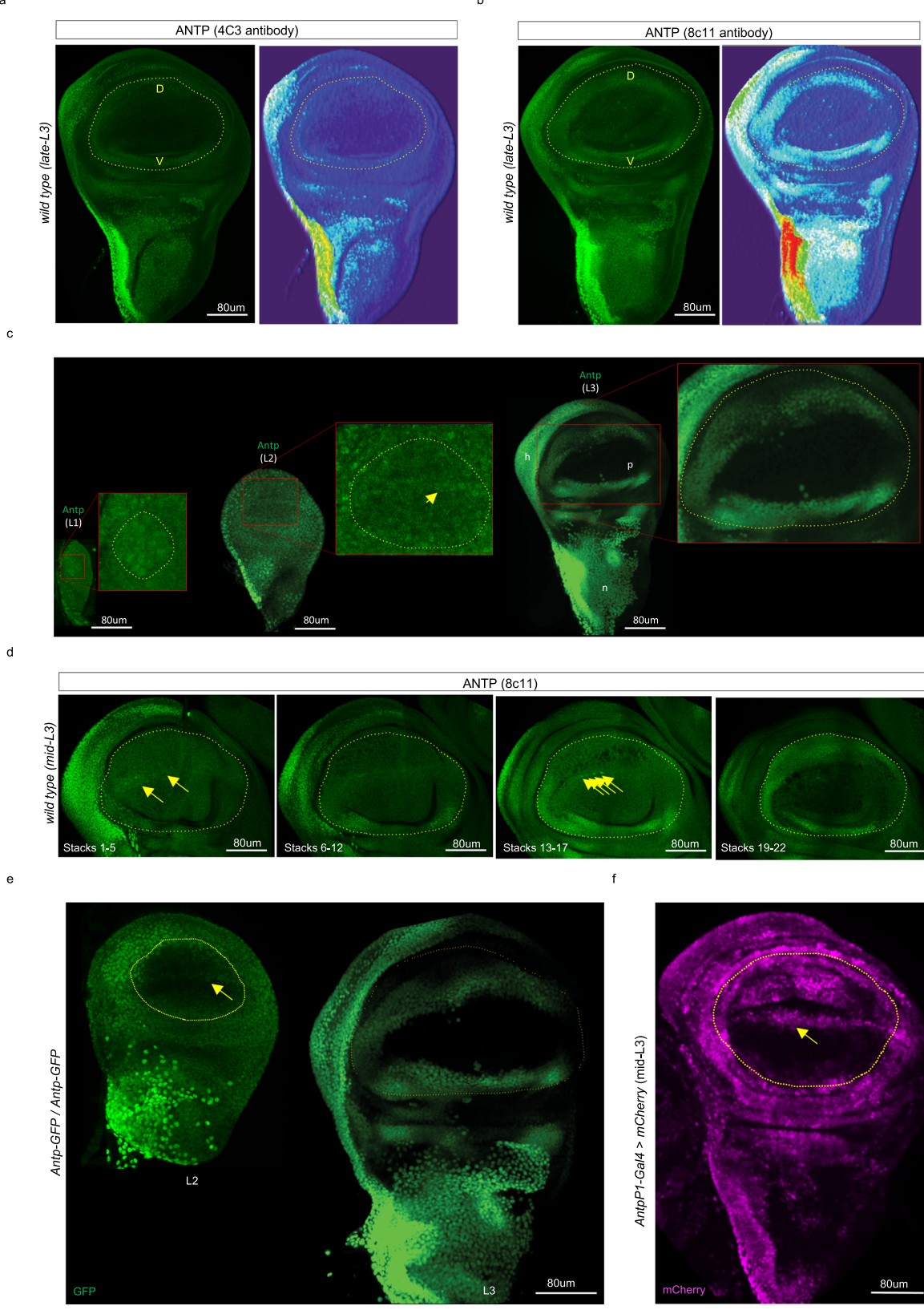

comparison, Antp is only expressed in the peripodial membrane and at very low level in a few cells of the hinge part in the haltere imaginal disc (Fig. 5a). Interestingly, the four halteres fly obtained with the *Cbx* mutation results from the loss of Antp expression in the wing disc and the gain of Ubx with a T3-like expression level in the pouch of the transformed disc (Fig. 5b, d). Conversely, the *Ubx* mutant background leading to the four-winged fly exhibited the de novo wing-like expression pattern of Antp and a strong reduction of Ubx expression in the pouch of the T3 haltere-to-wing transformed disc (Fig. 5c, d). Together, these observations underlined that the particular expression of Antp in the wing disc pouch is specifically linked to the wing developmental program.

**Fig. 1 The Hox protein Antennapedia (Antp) is dynamically expressed in the pouch of the *Drosophila* wing disc. a** Antp staining (green, revealed with secondary anti-mouse coupled to Alexa488) with the historical 4C3 antibody used for the first description of Antp expression in the wing imaginal disc[5]. **b** Antp staining (green, revealed with secondary anti-mouse coupled to Alexa488) with the 8C11 antibody used in this study. The pouch region is outlined (yellow dotted line). Pictures on the right are 3D-plot projection made with Fiji to better highlight the zones with high and low levels of expression (the low-to-high expression level follows the cyan–white–green–yellow–red color gradient). The dorsal (D) and ventral (V) wing blade compartments are indicated. **c** Expression profile of Antp (green) in the wing disc at the larval stages 1 (L1), 2 (L2), and 3 (L3). The pouch region is outlined (yellow dotted line) and enlarged (red box) to better show the typical expression pattern that becomes progressively restricted to the ventral and dorsal sides. Yellow arrow depicts the peculiar profile in cells across the dorsal–ventral boundary of the pouch, which is apparent from L2 to mid-L3 larval stages. The pouch (p), hinge (h), and notum (n) regions are indicated in the L3 wing disc. This expression profile was observed in four independent experiments ($n = 25$). **d** Illustrative Antp staining (green) across different stacks (each stack is 0.2 μm) of a confocal acquisition of mid-L3 wing disc. Note the homogenous expression in the dorsal and ventral sides of the pouch (yellow dotted line) and the line of low- and few high-expressing cells along the dorsal–ventral boundary of the pouch (yellow arrows). **e** GFP staining (green) of the mimic-modified line homozygous for endogenous *Antp-GFP*. A L2 and L3 wing disc is shown. The pouch is outlined (yellow dotted line). **f** Expression of the *UAS-mCherry* reporter (magenta) driven by the *P1-Antp* promoter in front of *Gal4* in mid-L3 wing disc. This driver shows more intense staining across the dorsal–ventral boundary of the pouch (yellow arrow). GFP and mCherry profiles were observed in two independent experiments (with a minimum $n = 10$).

To better understand the specific input of Antp on flight appendage development in *Drosophila*, we tested whether *Antp* could directly substitute *Ubx* and rescue the four-wings *Ubx* mutant phenotype. Intriguingly, similar to Ubx[23], high levels of Antp expression were sufficient to control/drive haltere formation (Fig. 5e, f-f′). The Hox gene *Abdominal-A* (*Abd-A*) was also able to rescue the four-wings *Ubx* mutant phenotype (Fig. 5e, f-f′), as previously described[23], and in accordance with the high degree of molecular and functional similarity between Ubx and Abd-A. In contrast, the more divergent Hox gene *Deformed* (*Dfd*) was not able to rescue the *Ubx* mutant phenotype, highlighting that not all Hox genes could substitute *Ubx* in this context (Fig. 5e, f-f′). In any case, the haltere rescue assays indicate that haltere formation is tightly linked to a high dose of the Hox gene product.

We, therefore, compared more precisely the expression level of Antp and Ubx in the wing and haltere disc pouch. Immunostaining with Antp and Ubx antibodies revealed that Ubx was 18.5 times more strongly expressed than Antp (Fig. 6a-a′). Because the affinity could be different between the two antibodies, we also used the *Antp-GFP* and *Ubx-YFP* lines for quantifying the expression level with the same anti-GFP antibody. This analysis revealed that Ubx expression levels were 10.5 times higher than Antp levels (Fig. 6b-b′), confirming that the wing and haltere developmental programs are linked to distinct Hox doses. To further evaluate the impact of the Hox dose on the wing and haltere development, we gradually modified the Hox expression level in the pouch of the wing and haltere discs. These genetic experiments showed that a gradual increase of Antp level in the wing pouch led to a progressive decrease of the wing size (Fig. 7a-a′). Importantly, the wing resembled a haltere in terms of size, shape, and hairs organization when the expression level corresponded to 490% of endogenous Antp (five times higher than endogenous Antp, Fig. 7a-a′). Conversely, a gradual loss of Ubx led to a progressive increase of the haltere size, and characteristic wing-like phenotypic traits (including veins, hairs organization, and flattening) eventually appeared when the level decreased to 39% of endogenous Ubx (2.5 times lower than endogenous Ubx, Fig. 7b-b′). In both cases, intermediate levels led to intermediary phenotypes. In combination with the rescue experiment, these results demonstrated that the formation of wing or haltere is not dependent on a specific Hox protein (here Antp or Ubx), but on a specific Hox dose in *Drosophila*.

**Antp and Ubx are expressed in the flight primordia of four-winged insect species.** To assess whether the Hox-free state model could be generalized or not among the insect kingdom, we analyzed *Antp* and *Ubx* expression in the flight appendage primordia of four-winged insect species of the orders Odonata,

Hymenoptera, and Lepidoptera, covering <350 million years of insect evolutionary time[24]. Insect species were chosen according to their flight appendage formula on the T2 and T3 segments, exhibiting either similar (the damselfly *Ischnura elegans*: Fig. 8) or different (the bee *Apis mellifera* with smaller HWs and the silkworm *Bombyx mori* with FW and HW of different shape: Fig. 8) wing morphologies. Immunofluorescence assays with Antp and Ubx cross-species antibodies revealed Antp nuclear staining in both the FW and HW primordia in all studied insect species, whereas Ubx nuclear staining was mostly restricted to the HW primordium (Fig. 8). Quantitative real-time PCR (RT-qPCR) experiments were also performed to quantify *Antp* and *Ubx* mRNA expression level in each flight appendage primordium (see "Methods" and Supplementary Figs. 4–6). Results showed that the overall *Antp* + *Ubx* mRNAs expression level was similar between the FW and HW primordium in *I. elegans* (with HW/FW ratio equal to 1.19: Fig. 8), but not in *A. mellifera* and *B. mori* (with respectively 2 times and 3.5 times higher Hox expression level in the HW primordium: Fig. 8). Thus, similar or dissimilar Hox expression level correlates with FW and HW of similar (in *I. elegans*) or dissimilar (different size and shape in *A. mellifera* or only different shape in *B. mori*) morphology in the adult. This observation suggests that the Hox dosage could be instructive in these insect species.

**Discussion**

Our results in *Drosophila* challenge the Hox-free paradigm of wing formation, with the description of an early and dynamic expression profile of Antp that correlates with a functional requirement for correct wing margin and wing size formation in the adult. This role applies from L1 to L3 stages, suggesting that Antp could probably act at different levels for regulating the cascade of downstream target genes across larval development, as described for Ubx[1,25]. The role of Antp in the wing disc pouch is also linked to a weak expression level when comparing to its expression level in the hinge or notum region, and importantly, when comparing to the Ubx expression level in the haltere disc pouch. Interestingly, the Hox cofactor Hth acts as an upstream regulator to either activate or repress Antp (this study) or Ubx[22] in the wing and haltere disc pouch, respectively. We further showed that this differential Hox expression level is important for controlling the wing and haltere fate, highlighting that the formation of these two flight organs is directly dependent on the Hox dosage and not on the Hox type (at least here Antp and Ubx). We also observed a particular Antp expression profile in the T2 wing disc pouch, which was found in the haltere-to-wing transformed tissue disc of *Ubx* mutant individuals. Interestingly, RNAi experiments with *MS1096* led to residual Ubx expression along the DV

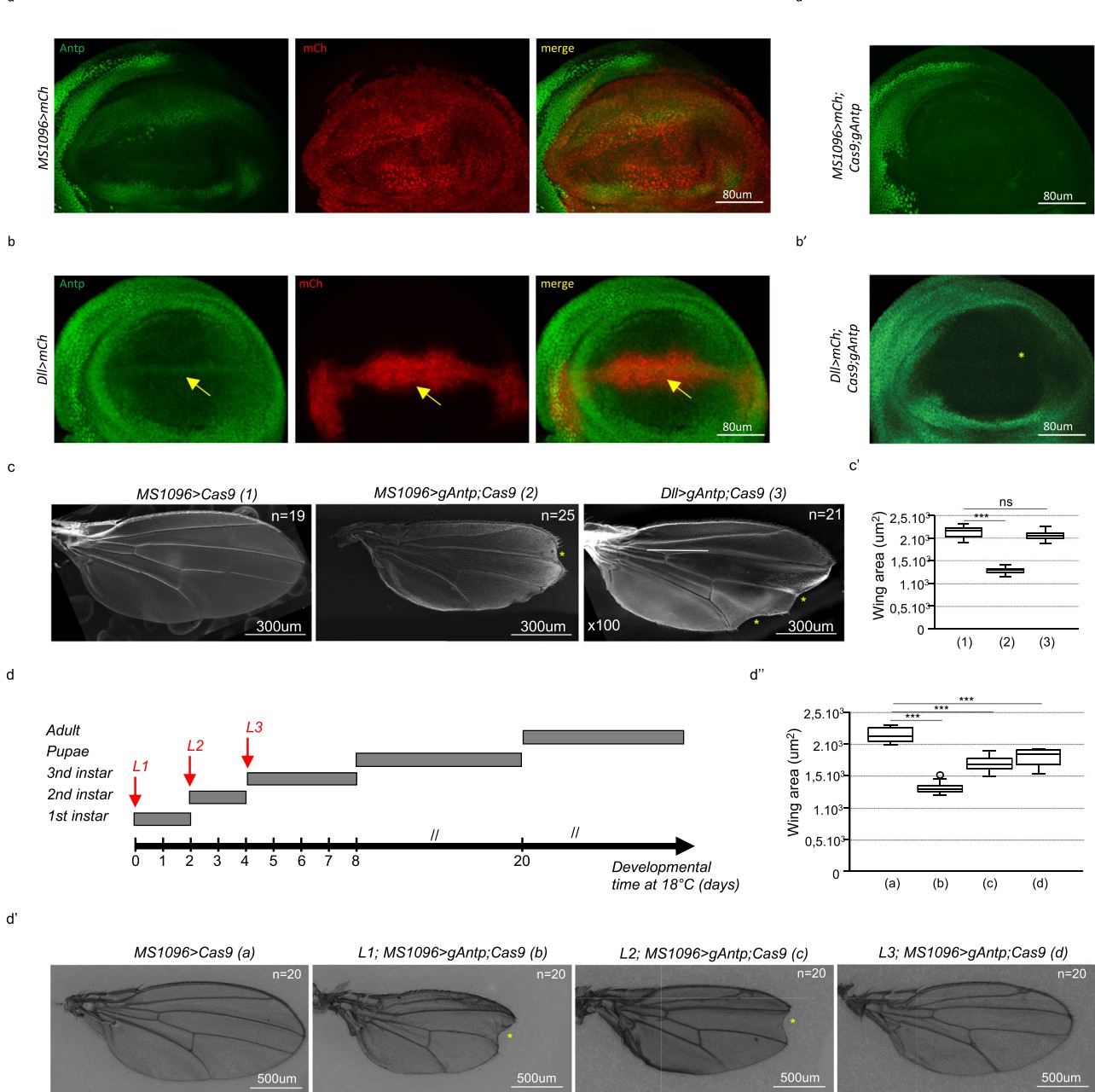

**Fig. 2 Antennapedia (Antp) is required for correct distal wing formation in Drosophila. a, b** Expression of the red fluorescent reporter mCherry (mCh, red) with the *MS1096-Gal4* (**a**) or *Distalless(Dll)-Gal4* (**b**) driver does not affect the expression of Antp (green) in the wing pouch (*n* = 5 from one experiment). Yellow arrow highlights *Dll-Gal4* and Antp co-expressing cells along the dorsal–ventral boundary of the early L3 wing disc pouch. **a′, b′** Expression of *gRNAs* and Crispr/Cas9 against *Antp* with *MS1096* (**a′**) or *Dll-Gal4* (**b′**) abolished the expression of Antp (green; *n* = 10 from two independent experiments). Note that *MS1096* starts at the L1 stage and its expression profile is not restricted to the pouch. **c** Illustrative SEM pictures of adult wings from control (*MS1096-Gal4* driver with *UAS-Cas9*, *n* = 18) and *Antp*-mutant (*MS1096-Gal4* or *Dll-Gal4* driver with *UAS-Cas9* and *UAS-gRNAs* against *Antp*, *n* = 25 or 19, respectively) males, as indicated. The yellow star indicates wing margin defect (observed in 61 and 90% of *MS1096-Gal4* and *Dll-Gal4* dissected wings, respectively). **c′** Boxplot representation of wing area quantification in the control (1) and *MS1096-Gal4* (2) or *Dll-Gal4* (3) induced *Antp*-mutant flies. Boxplots indicate 25th and 75th percentiles, whiskers show ±1.5 × IQR and center line depicts median of three biological replicates (unpaired two-tailed *t* test with Welch's correction ***p = 1.66e−20 with *MS1096* and nonsignificant (ns) with *Dll-Gal4*. **d** Developmental time course of *Drosophila* larval stages at 18 °C. Shifts at 29 °C to abolish the activity of Gal80 and induce the expression of *Antp gRNAs* were performed at early L1, L2, and L3 stages, as indicated. **d′** Illustrative pictures of adult wings in the control experiment (**a**) or upon the expression of *Antp gRNAs* with the *MS1096-Gal4* driver at the L1 (**b**), L2 (**c**), and L3 (**d**) stages. Margin defects (yellow stars) were observed when *Antp* expression was affected at the L1 (58%) and L2 (43%) but not L3 stage. **d″** Quantification of wing size defects in the different conditions. Boxplots indicate 25th and 75th percentiles, whiskers show ±1.5 × IQR and center line depicts median of three biological replicates (unpaired two-tailed *t* test with Welch's correction ***p = 1.8e−24 at L1, ***p = 6.5e−15 at L2, ***p = 1.2e−10 at L3).

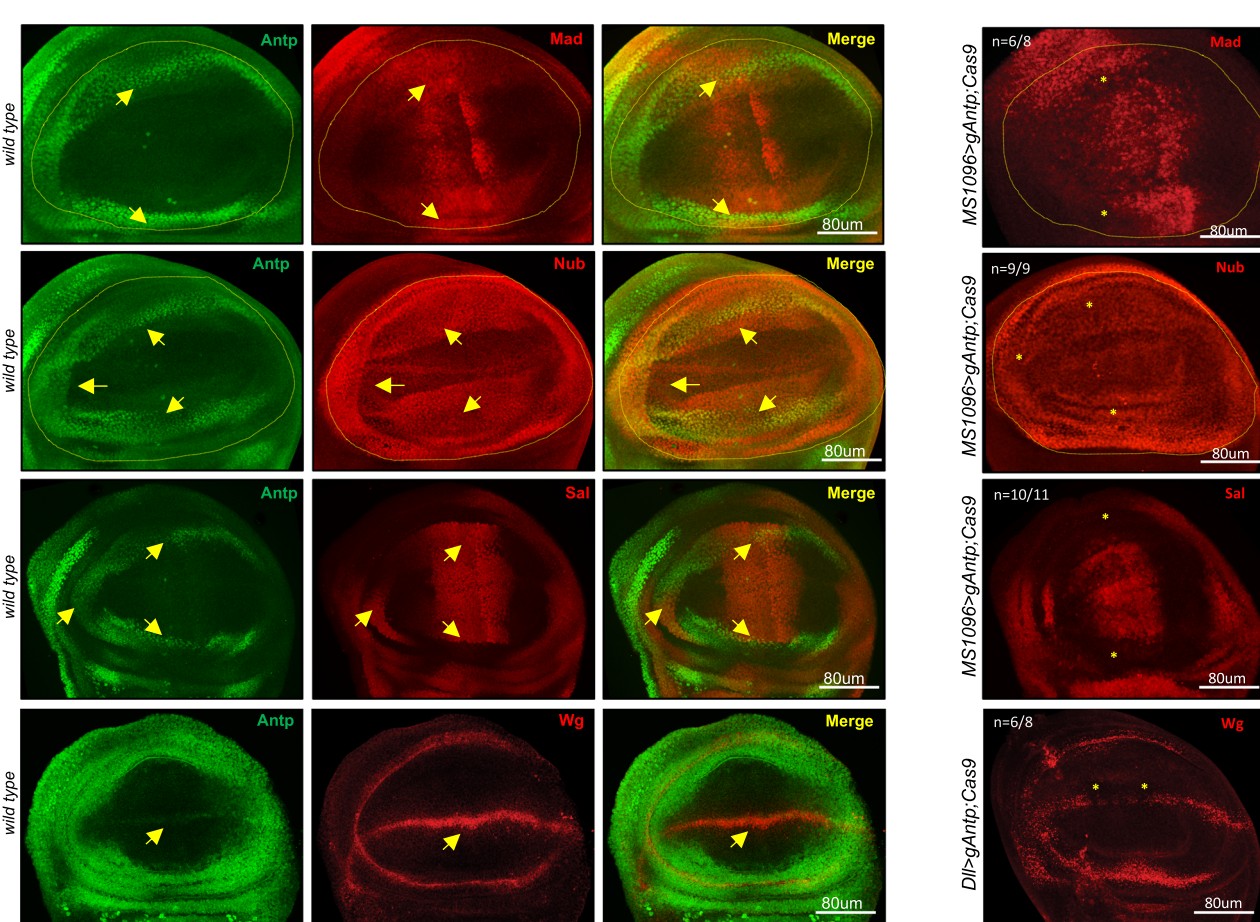

**Fig. 3 Loss of Antennapedia (Antp) affects genes involved in wing formation. a** Expression of Antp (green) and genes involved in the wing formation (red) in the wild-type wing disc pouch (yellow dotted line), as indicated. Overlapping expression regions are indicated (yellow arrows). **a'** Expression of Antp (green) and genes involved in the wing formation (red) in the *Crispr/cas9*-mediated *Antp*-mutant wing disc pouch (yellow dotted line), as indicated. Wing-specifying genes are specifically lost in the overlapping expression regions with Antp (yellow stars). The number of mutant discs showing the same pattern over the total number of dissected discs (from two independent experiments) is indicated.

boundary, recalling a spatial pattern that resembled to Antp in the wing disc. These two observations suggest that the distinct roles of Antp and Ubx in the wing and haltere discs are likely linked to distinct spatial expression profiles in addition to specific doses. Along this line, discrete high expression domains of Antp have been described to be associated with the formation of pigmentation eyespots in the FW and HW of butterflies[9], underlining that specific Hox expression profiles could be involved in the control of various phenotypic traits in insect wings.

Analyses in four-winged insect species further confirmed that the expression of Antp in the wing primordium is not specific to *Drosophila*. Antp was even found to be systematically expressed in both the T2 and T3 flight primordia in all the studied species. In addition, the overall quantification of Antp and Ubx expression level in the two flight primordia showed a striking correlation with the presence of distinct or similar pairs of wings on the T2 and T3 segments. This correlation is evident at the level of wing size in the damselfly and bee, while the distinct expression level observed in *B. mori* correlates with different wing shapes. These various impacts could potentially be due to various temporal requirements, as observed in our temperature restrictive assays in *Drosophila*. For practical reasons, quantifications have been performed at late stages in the different species, and this aspect could therefore not properly be addressed. In addition, these stages are not equivalent and the expression patterns and

levels cannot be compared between the different species (the Q-PCR conditions are not identical and the affinity of Antp and Ubx antibodies is not known in the different species). Still, the observation that Antp could be expressed in addition to Ubx in the HW primordium of four-winged species and not in *Drosophila* suggests that wing developmental programs could be differently sensitive to the Hox dose in different insect species. In any case, although our observations in four-winged species need further genetic validation, we propose a speculative model where variation in the Hox dosage (and possibly the spatial profile) could be used to diversify flight appendage morphology, from subtle wing size and/or shape modification to the formation of a completely different organ during insect evolution (Fig. 9).

The direct impact of the Hox dose on morphological diversification in animals remains surprisingly poorly investigated, and only two examples are known, the control of the size and number of digits in vertebrates[26] and the leg length in a water strider species[27]. Previous work described the importance of auto-regulation to buffer against increases in Ubx protein level[28] and the fine-tuning of *Ubx* transcription in a compartment-specific manner within the haltere imaginal disc[22]. Interestingly, recent studies showed that the nutritional status could control the level of Ubx in a rice planthopper insect, with long and short wings linked to a low and high level of expression, respectively[29,30]. In this study, we identify the importance of the Hox dosage in the

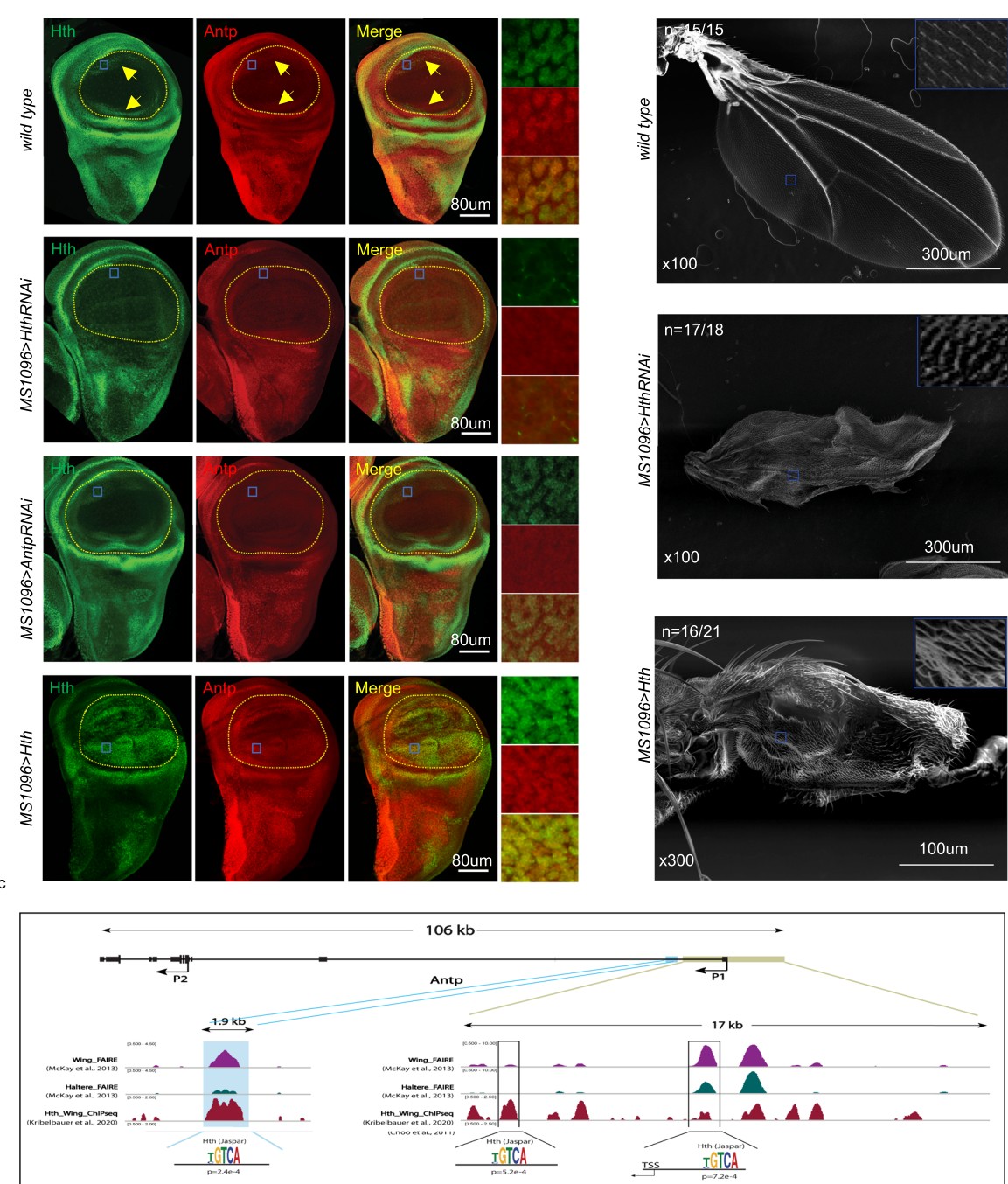

**Fig. 4 Homothorax (Hth) is a positive regulator of *Antennapedia* (*Antp*) in the dorsal and ventral parts of the wing disc pouch. a** Expression of Hth (green) and Antp (red) in the different genetic contexts, as indicated. Yellow arrows highlight the dorsal and ventral part of the pouch where Hth and Antp overlap. Enlargement on co-expressing nuclei (scare) is also indicated on the right. The pouch in wild type or domain of expression of the *MS1096* driver is outlined (yellow dotted line). These patterns were systematically observed in dissected discs from two independent experiments (minimum *n* = 8). The enlargement in the *MS1096 > UAS-Hth* background is shown from a region of the pouch, where Antp is normally not expressed. **b** SEM acquisition of wild-type and mutant adult wing, as indicated. The number of wings showing the same phenotype over the total number of dissected wings from adult males (from two independent experiments) is indicated. **c** Hth-binding profile on the *P1 cis*-regulatory sequences of *Antp* from Chip-seq performed in the wing disc, as indicated. Several Hth-binding sites are found in wing-specific chromatin-accessible regions (as deduced from FAIRE analysis and in comparison, to the haltere disc).

course of wing evolution both by expression analyses in several insect species and genetic arguments in *Drosophila*. We propose that Hox dosage variation is an underestimated yet likely widely used molecular strategy to diversify Hox activity and thereby morphologies during animal evolution.

## Methods

**Drosophila strains**. *Drosophila* strains were cultured following standard procedures at 25 °C. *Yellow white* was used as a wild-type strain. Hox constructs fused to the C-terminal fragment of Venus (VC) have previously been used[31]. *Antp-GFP* fly line was generated by using the corresponding MiMIC fly line (*Antp^{MI02272}*, line 33187 at the Bloomington stock center) for inserting the GFP-encoding fragment

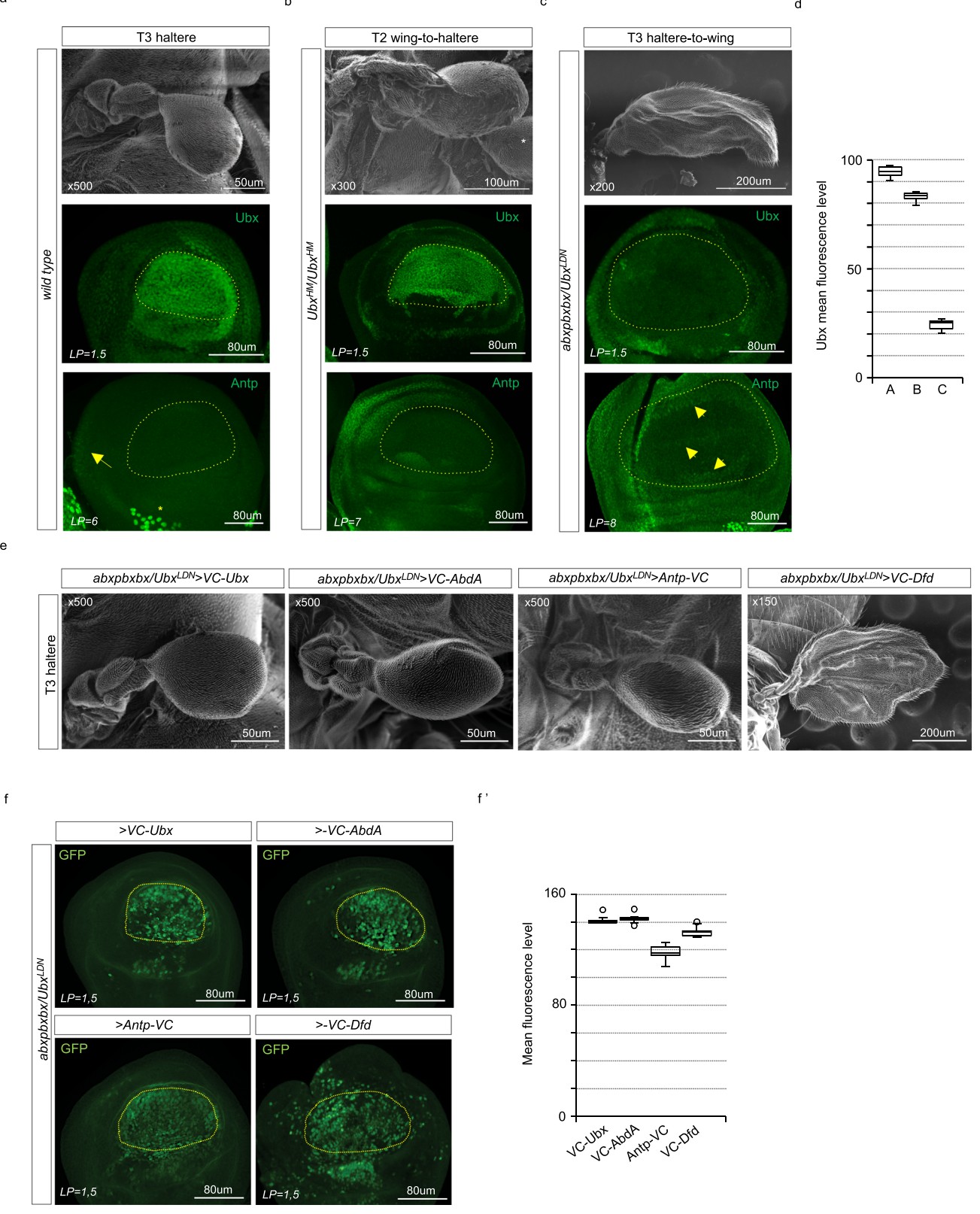

(MiMIC injection service of Bestgene Inc.)[32]. The *Ubx-YFP* fly line is from the Cambridge protein trap project Flyprot (line CPTI-000601). Suitable gRNAs targeting *Antp* were designed to bind in the first exon, and were identified using the webtool CRISPR Optimal Target Finder (http://targetfinder.flycrispr.neuro.brown.edu). Four different gRNAs were cloned together into the pCFD6 plasmid, according to the protocol published on https://www.crisprflydesign.org (*Antp-gRNA1*:GATGACGCTGCCCCATCACA, *Antp-gRNA2*:GGCCGTTGTAG TAGGGCATG, *Antp-gRNA3*:GGCGGGATCAGCAGACGCTG, and *Antp-gRNA4*:

GGTTCTGATGGACCTGTGAT). The *pCFD6-UAS-Antp-gRNAs* construct was injected by BestGene (attP5, third chromosome)[33]. Primers and sequence maps are available upon request. The following GAL4, Gal80ts, and UAS stocks were used: *Ubx-GAL4^{LDN}*[34], *Ubx^{HM1}*, *abxpbxbx*[34], *Nab-Gal4* (ref. [1]), *Dll-Gal4* (ref. [35]), *MS1096-Gal4* (Bloomington, #8696), *MS209-Gal4* (Bloomington, #25676), *AntpP1-Gal4* (Bloomington, #26817), *Nubbin-Gal4* (Bloomington, #25754), *UAS::hthRNAi* (*hth^{KK108831}* line from VDRC), *UAS::AntpRNAi* (Bloomington #27675), *UAS::UbxRNAi* (*Ubx^{GD5049}* from VDRC), and *Tubulin-Gal80ts* (Bloomington, #7018).

**Fig. 5 Antennapedia (Antp) can rescue the *Ultrabithorax* (*Ubx*)-mutant haltere phenotype upon Ubx-like expression profiles. a** Illustrative SEM picture of adult wild-type haltere and expression of Ubx or Antp (green) in the wild-type haltere disc. Antp is weakly expressed in a few cells of the hinge (yellow arrow) and in surrounding mesodermal cells (yellow star). **b** Illustrative SEM picture and expression of Ubx or Antp (green) in the T2 wing-to-haltere transformed tissue in *Ubx^HM* homozygous individuals. 100% of individuals displayed the same phenotype. The wing-to-haltere transformation correlates with the loss of Antp and the gain of Ubx expression in the pouch. White star in the SEM picture indicates the normal posterior T3 haltere. **c** Illustrative SEM picture and expression of Ubx or Antp (green) in the T3 haltere-to-wing transformed disc of *Ubx^LDN/abxpbxbx* individuals. 100% of individuals displayed the same phenotype. Note the strong decrease of Ubx (within and outside the pouch) and the gain of Antp expression in the pouch (yellow arrows), which is reminiscent of the normal Antp profile in the wing pouch. The pouch region is outlined in the wing and haltere disc (yellow dotted line). Laser power (LP) is indicated. **d** Boxplot representation of the quantification of Ubx expression level in the pouch in the different genetic backgrounds (A: *wild type*; B: *Ubx^HM*; and C: *Ubx^LDN/abxpbxbx*). Boxplots indicate 25th and 75th percentiles, whiskers show ±1.5 × IQR and center line depicts median of two biological replicates, (minimum *n* = 15). **e** Illustrative SEM pictures of haltere rescue in the *abxpbxbx/UbxGal4^LDN* background upon the exogenous expression of VC-Ubx (*n* = 16/17), VC-Abd-A (*n* = 15/17), Antp-VC (*n* = 12/17), and VC-Dfd (*n* = 10/10). **f** Immunostaining of the VC-Hox fusion proteins expressed in the *abxpbxbx/UbxGal4^LDN* background with the GFP antibody recognizing the VC fragment[40]. The pouch region is outlined (yellow dotted line). **f'** Boxplot representation of the quantification of the expression level of Hox fusion constructs in the haltere pouch, as deduced from the GFP immunostaining. Boxplots indicate 25th and 75th percentiles, whiskers show ±1.5 × IQR and center line depicts median of two biological replicates (minimum *n* = 10 discs). Laser power (LP) is indicated.

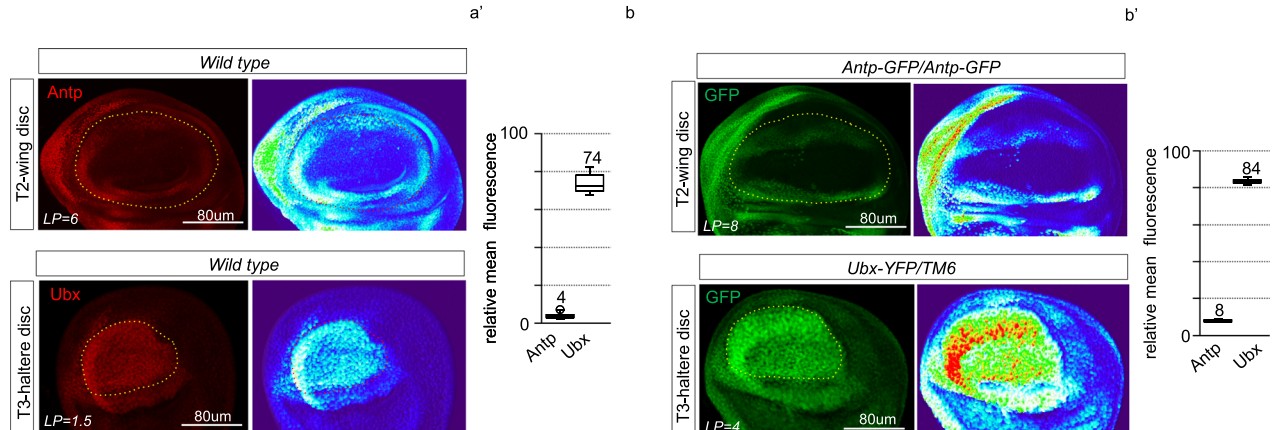

**Fig. 6 Quantification of Antennapedia (Antp) and Ultrabithorax (Ubx) expression levels in the wing and haltere pouch, respectively. a** Antp and Ubx immunostaining (red, revealed with secondary anti-mouse coupled to Alexa555) in the wing or haltere pouch of late L3 larva, respectively. A 3D surface plot profile is shown for each staining to better highlight the various levels of expression (using the Fiji thermal LUT color code). **a'** Boxplot representation of the quantification of the mean fluorescence intensity when considering all Antp and Ubx expressing cells in the wing and haltere pouch, respectively. Boxes indicate median and 25th and 75th percentiles. Boxplots indicate 25th and 75th percentiles, whiskers show ±1.5 × IQR and center line depicts median of two biological replicates (*n* = 20 discs). The final quantification takes into account the laser power (LP) used for confocal acquisitions (which is four times less strong for Ubx immunostaining). The ratio between Ubx and Antp immunostaining (74/4) shows a fold change of 18.5. **b** Illustrative confocal picture of GFP staining (green) in the T2 wing disc or T3 haltere disc of homozygous *Antp-GFP* or heterozygous *Ubx-YFP* individuals, respectively. A 3D surface plot profile is shown for each staining. **b'** Boxplot representation of the quantification of the mean fluorescence intensity when considering all GFP-positive cells in the wing and haltere pouch, respectively. Boxplots indicate 25th and 75th percentiles, whiskers show ±1.5 × IQR and center line depicts median of two biological replicates (*n* = 20 discs). The final quantification takes into account the laser power (LP) used for confocal acquisitions (which is two times less strong for GFP immunostaining in the haltere pouch) and the genotype (two copies for *Antp-GFP* and one copy for *Ubx-YFP*). The ratio between the haltere and wing disc pouch (84/8) shows that Ubx is 10.5 times more highly expressed than Antp. Considering that the GFP antibody has the same affinity in the wing and haltere disc pouch, we conclude that the Antp antibody is 1.85 times less affine than the Ubx antibody.

**Immunofluorescence assay in *Drosophila* imaginal discs.** Imaginal discs were fixed following dissection in 4% paraformaldehyde (methanol free) for 15 min. Washes were done with 1× PBS 0.1%TritonX solution (PBTx). Samples were then blocked with 2% BSA solution for 2 h. Primary antibodies were incubated for 24 h at 4 °C, and then washed in PBTx and secondary antibodies incubated for 2 h at room temperature. Samples were then washed in PBTx and mounted in Vectashield (Vector laboratories) for confocal acquisition. Primary antibodies used were mouse anti-Antp 4C3 (1:20; DSHB); mouse anti-Antp 8C11 (non-diluted; DSHB); mouse anti-Ubx FP3.38 (1:200; DSHB); mouse anti-Ubx/ABD-A FP6.87 (1:20; DSHB); rabbit anti-GFP PABG1 (1:500; Chromotek); rabbit anti-PDM1 (nubbin, 1:500; from J. Enriquez lab); mouse anti-Wg 4D4 (1:200; DSHB); and rabbit anti-Spalt (1:200) and anti-Hth (1:200) antibodies (from J. Enriquez lab).

**Immunofluorescence assay in *Ischnura*, *Apis*, and *Bombyx* flight primordia/imaginal discs.** Wing primordia (*I. elegans* and *A. mellifera*) or imaginal discs (*B. mori*) were fixed following dissection in 4% paraformaldehyde (methanol free) for 15 min. Washes were done with 1× PBS 0.3%TritonX solution (PBTx). Samples were then blocked with 2% BSA solution for 24 h. Primary antibodies were incubated for 48 h at 4 °C and then washed in PBTx and secondary antibodies

incubated for 24 h at 4 °C. Samples were then washed in PBTx and mounted in Vectashield (Vector laboratories) for confocal acquisition.

**Microscopy and imaging.** All the fluorescence microscopy images of imaginal discs and wing primordia were captured using confocal Leica SP8. Images were recorded at a 1024 × 1024-pixel resolution using oil objective 40×. Expression intensity of GFP, Antp, and Ubx was determined using the histogram function of the FIJI Software. Briefly, threshold was adjusted (using the « Image calculator » function) to create an image containing all positive nuclei (using the « Substract » function) that were analyzed for fluorescence quantification (using the « analyze particles » function) and deduce the mean fluorescence intensity. The adult *Drosophila* appendage phenotype images were taken by scanning electron microscope Hirox SH-3000 or with a Keyence VHX7000 microscope. The adult wing and haltere were isolated from the rest of the insect body to allow for better manipulation, while mounting samples. Picture of other adult insects were taken with a Keyence VHX7000 microscope.

**Quantitative real-time PCR**
*Tissue, RNA collection, and reverse transcription (RT).* Tissues were collected either from larvae (*B. mori*) or from nymphs (*A. mellifera* and *I. elegans*). T2 and T3 wing

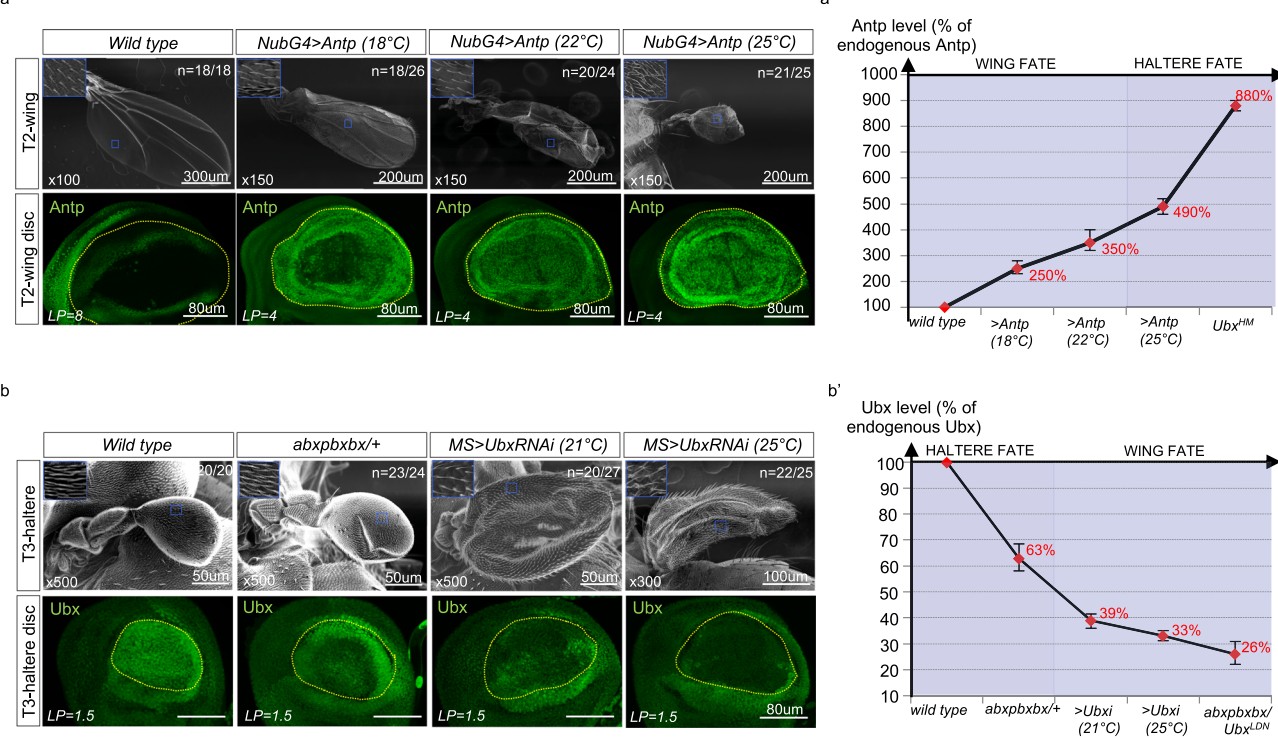

**Fig. 7 Different Hox doses are instructive for flight appendage morphology in *Drosophila*. a** Increasing the dose of *Antp* in the wing disc pouch led to progressive wing-to-haltere transformation phenotypes. Overexpression of Antp was performed with the *Nub-Gal4* driver at different temperatures. This driver starts to be expressed at late second instar larval stage[44]. Effects were observed in two independent experiments and fluctuate from 69% (18 °C, *n* = 18/26) to 83% (22 °C, *n* = 20/24) and 84% (25 °C, *n* = 21/25) of the total number of dissected wings. Enlargements on SEM pictures show hairs' shape and organization, which look like in the haltere at the highest Antp dose. Bottom panels show the immunostaining of Antp in each condition (green, a minimum *n* = 10 from two independent experiments). **b** Decreasing the dose of *Ubx* in the haltere disc pouch led to progressive haltere-to-wing transformations. Ubx expression was affected by using the *abxpbxbx* heterozygous allele, and by expressing RNAi against *Ubx* with the *MS1096-Gal4* driver at two different temperatures. These effects were robustly observed in two independent experiments and fluctuate from 96% (*abxpbxbx/+*, *n* = 23/24) to 74% (21 °C, *n* = 20/27) and 88% (25 °C, *n* = 22/25) of the total number of dissected halteres. Enlargements on SEM pictures show hairs' shape and organization, which look like in the wing at the lowest Ubx doses. Bottom panels show the immunostaining against Ubx in each condition (green, a minimum *n* = 10 from two independent experiments). **a′, b′** Curves showing the relationship between the wing or haltere fate and the level of Antp or Ubx (given as a percentage of the corresponding endogenous expression level in the wild-type condition, and deduced from the mean quantification relative to the laser power (LP) for fluorescent immunostaining in the pouch). Wing and haltere fates were deduced from SEM acquisitions, taking into account the size, the presence or not of veins, and the organization of hairs. Bars represent mean ± SD of two independent experiments.

imaginal discs or wing primordia were collected separately, and dissections from three to four individuals were pulled in each sample. RNA was extracted using NucleoZol (Macherey-Nagel). RNA concentration was evaluated using Qubit4 fluorometer (Thermo Fisher). RT was carried out 5 min at 70 °C followed by 1 h at 42 °C in 20 µl reactions containing 0.5 mM dNTP each, 10 mM DTT, 0.5 µg oligo (dT)15 (Promega), and 200 U of M-MLV-Reverse Transcriptase (Promega).

*Primers design and real-time qPCR.* All primers were designed to have comparable biochemical properties (%GC content and melting temperature). For Antp and Ubx, primers were designed within the highly conserved homeodomain (HD) common to most different transcripts to avoid excluding any transcripts with putative transcriptional activities (Supplementary Figs. 4 and 5). Because Ubx sequence is not identified in *I. elegans*, we used a low-stringency annealing temperature to amplify its HD with primers derived from the HD of *Limnoporus dissortis*. The amplified fragment was separated in a 2% agarose gel, cloned into pCR II TOPO vector using the TOPO TA cloning kit (Thermo Fisher Scientific) and sequenced. The obtained Ubx sequence for *I. elegans* undoubtedly match those of the Ubx HD in the other studied species and was used to design new specific primers for the qPCR.

RT-PCR was conducted in a 10 µl reaction containing 0,1% of RT product, 2 µM of each dNTP, 10 pmol of each primer, and 5 µl of Sybergreen mix (iTaq Universal SYBR Green Supermix Biorad). For each set of primers, amplification efficiency was first tested using a gradient of annealing temperatures in order to harmonize annealing temperature for all sets. For all PCR reactions, cDNA was denatured 10 min at 95 °C and amplified for 40 cycles in a two steps program as following: 30 s at 95° followed by 30 s annealing and polymerization at 65°. For each set of primers, standard curves were performed using serial dilution of the amplicon based on the calculation of the molecular weight specific to each amplicon. Finally, standard curves were used to calculate mRNA copy numbers in the analyzed samples (expanded view

2). The amount of template was adjusted to equal quantity between samples within the same tissue of the same species based on the level of Tubulin (Tub) for *A. mellifera* and *B. mori*, or the level of Actin (Act) for *I. elegans*.

**Chip-seq analyses.** No new datasets were generated in this study. Fastq files for Hth ChIP-seq in the wing imaginal discs were downloaded from NCBI GEO datasets (GSM3578084 and GSM3578085) and aligned to the dm6 assembly, using the bwa-mem software[36]. Sam files were further converted to bam, duplicate reads removed, and sorted using the samtools package, version 1.9 (Li H, 2009). The bamCompare function from the Deeptools package (version 3.1.3, ref. [37]) was used to generate the input normalized bigwig file. The FAIRE seq wiggle files for the haltere and wing imaginal discs were obtained from the NCBI GEO datasets (GSM948716 and GSM948717). The reference assembly was modified to dm6 version using the CrossMap software[38]. All alignments were visualized using the IGV software (version 2.8.0)[39] and Hth-binding site prediction was performed, using the MAST program (version 5.1.1)[40] from the MEME suite.

**Single-cell RNA-seq data analysis.** No new datasets were generated in this study. Feature-barcode matrices was retrieved from the NCBI Gene Expression Omnibus repository under the accession numbers GSM3902311. Further analysis was performed using the Seurat R package v3.1 (refs. [41,42]). We first normalized the data. Next, the data were scaled. Using the FindVariableFeatures function, 3000 features were identified as highly variable from cell-to-cell. We then performed linear dimensional reduction using the principal component (PC) analysis method, including 50 PCs and to visualize the data we used the uniform approximation and projection method[43]. We used the WhichCells function to select the *nubbin*-positive cells from the wing imaginal disc epithelial cells.

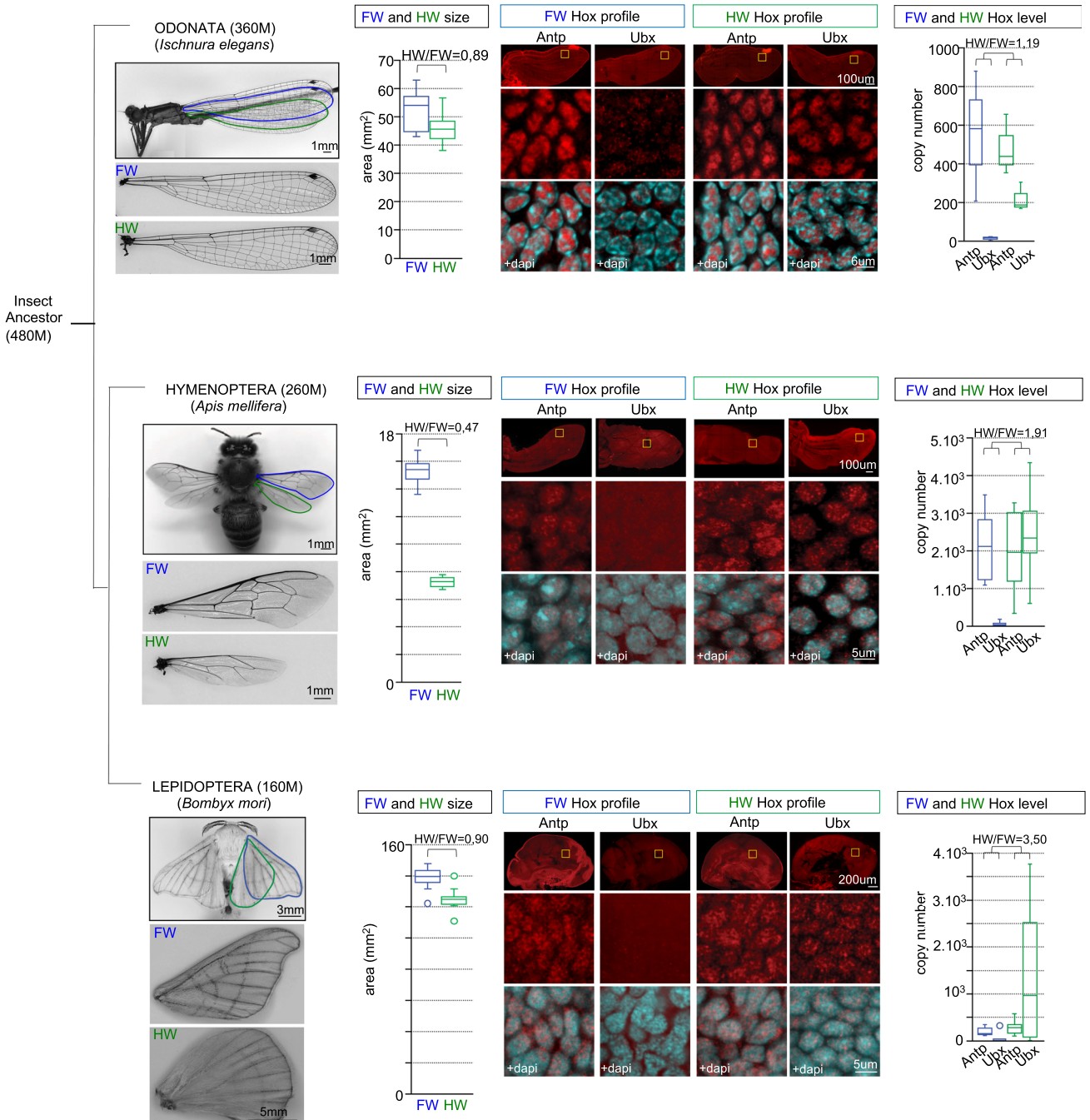

**Fig. 8 Antennapedia (Antp) and Ultrabithorax (Ubx) are expressed in the flight appendage primordia of evolutionary distant insect species.** Simplified insect evolutionary tree with illustrated pictures of insect species used in this study (the timescale is given in million (M) years). Individual forewings (FW, surrounded in blue) and hindwings (HW, surrounded in green) are shown below each adult species. Boxplots on the right show the quantification of the adult FW and HW area (minimum $n = 10$). The HW/FW ratio indicates the size difference between the two pairs of wings. Representative confocal images of the expression profile of Antp and Ubx (in red) in the full FW and HW primordium (top), and in enlargements (yellow box) on few cells to better highlight the specific nuclear staining (with nuclear DAPI co-staining in cyan). These patterns of fluorescent immunostaining were systematically reproduced from three independent experiments ($n = 8$ for *I. elegans*; $n = 10$ for *A. mellifera* and $n = 12$ for *B. mori*). Boxplots on the right show RT-qPCR quantification for *Antp* and *Ubx* mRNA from dissected FW and HW primordia of late nymph (*I. elegans*, $n = 12$) and pharate pupa (*A. mellifera*, $n = 18$) or from imaginal discs of fourth instar larva (*B. mori*, $n = 18$). The HW/FW ratio indicated above each graph is calculated by considering the total *Antp + Ubx* mRNA expression level in each primordium. Boxplots indicate 25th and 75th percentiles, whiskers show ±1.5 × IQR and center line depicts median of three biological replicates.

**Statistical information**. Statistical analyses were performed with Excel 2010 (Microsoft) and considered statistically significant at a *p* value < 0.05. When normality or equal variance between samples was achieved, an unpaired two-tailed *t* test was used. When normality or equal variance of samples failed, an unpaired two-tailed *t* test with Welch's correction was applied. All values are presented as mean ± SD. The number of experiments and samples are indicated in the figure legends. All pairwise comparisons were two-tailed. The investigators were not blinded to allocation during experiments and outcome assessment.

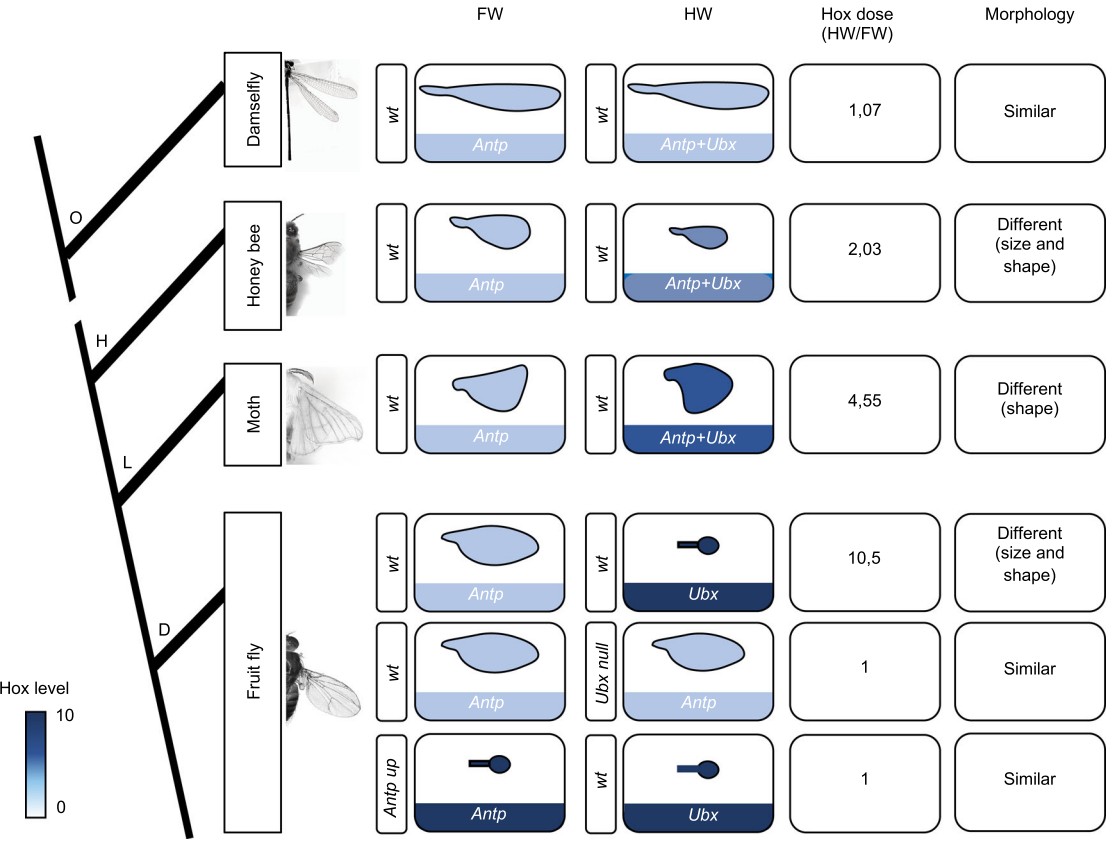

**Fig. 9 Speculative model on Hox dosage variation and the evolution of flight appendages in insects.** The model is based on the Hox (Antp or Antp + Ubx) expression level in the forewing (FW) and hindwing (HW) primordia. Ubx level was systematically considered as negligeable in the FW primordium of all studied species. Graded blue color depicts the Hox level (from light blue/low dose to dark blue/high dose). Note that although of similar colors, the doses are not comparable between the different species (RT-qPCR conditions and developmental stages were not identical). The total Hox dose when calculating the HW/FW ratio is indicated for each species, together with the corresponding morphological similarities or differences between the adult flight appendages. Note that morphological differences do not systematically correspond to size variation, as observed in *B. mori* that displays FW and HW of similar size, but of different shape. Genetic experiments in *Drosophila* showed that modifying the Hox dose by abolishing *Ubx* (which induces the derepression of Antp) or increasing Antp could respectively transform the haltere or the wing into a wing or a haltere. In both cases, similar Hox levels led to similar flight appendages on the T2 and T3 segments. The specific expression profile of Antp in the pouch of the wing disc or in the haltere-to-wing transformed disc (not schematized in the figure) also indicates that the spatial profile is likely not neutral in addition to the dose. O Odonata, H Hymenoptera, L Lepidoptera, D Diptera.

**Reporting summary**. Further information on research design is available in the Nature Research Reporting Summary linked to this article.

## Data availability

The data that support the findings of this study are available from the corresponding author upon reasonable request. Source data are provided with this paper.

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

## Acknowledgements

We thank the Bloomington and Vienna stock centers for providing the *Drosophila* fly lines, the Developmental Studies Hybridoma Bank for antibodies, and the Arthrotools platform of the UAR3444/US8 of Lyon for fly food. We thank Rosa Barrio for the Spalt antibody and Jonathan Enriquez for the Hth antibody. We thank M. Kmita and M. Averof for their helpful comments on the manuscript. Work in S. M.'s laboratory was supported by the CNRS, ENS-Lyon, Fondation pour la Recherche Médicale (FRM 160896), and Centre Franco-Indien pour la Promotion de la Recherche Avancée (Cefipra N°5503-P). Work in I.L.'s laboratory was supported by the DFG (LO 844/5-2).

## Author contributions

R.P.: conceptualization, execution of experiments, data analysis, formal analysis, and writing. G.G., K.D., M.D., and S.K.: execution of experiments and data analysis. S.V.: formal analysis. F.M.: execution of experiments, data analysis, and writing. I.L., R.S., and L.S.S.: writing. S.M.: conceptualization and data analysis, formal analysis and writing.

## Competing interests

The authors declare no competing interests.
