## [Peer Review File · Nature Communications]

Reviewers' Comments:

Reviewer #1:

Remarks to the Author:

Hox dosage is a rather over-looked aspect to Hox function and so the current manuscript is providing an important insight into these major developmental control genes. The authors also provide new expression data for the Antp protein in wing development of *Drosophila melanogaster*, which is impressive given the number of people that have studied this gene over a good number of years. This new expression data helps to reconcile some previously observed effects of Antp mutants in the wing development, which is satisfying.

I think that the results reported in this manuscript will be of widespread interest, to mechanistic developmental biologists, particularly those working on the Hox genes, as well as evolutionary developmental biologists, who will be pleased to see the inclusion of the inter-species comparisons made here. There are a wealth of new results reported in this manuscript and the experiments are carefully done and clearly documented.

My main comments are that the conclusions that can be drawn from these results might be a bit more nuanced than the authors currently describe.

1) The 'story' is not as simple as expression levels, as shown by Fig.1. Specifically, it is the *Bombyx mori* data that complicates matters. The damselfly data fits with the hypothesis that similar levels of Hox (Antp+Ubx) lead to similar sized wings, whilst *Apis mellifera* shows higher Hox levels (when Antp is summed with Ubx) in the hindwing correlating with a smaller wing size than the fore-wing. *Bombyx mori*, however, has much higher Hox levels in the HW, but not much of a size reduction of the HW. I missed a discussion of this apparently contradictory *Bombyx* result.

2) The ability of Antp to specify a haltere in T3 and to rescue haltere phenotype in the *abx-pbx-bx* mutants seems surprising, and perhaps counterintuitive (since Ubx is supposed to be the 'haltere-specifying' Hox gene). Does this mean that wing development requires very restricted Hox (Antp) expression, such as just along the wing boundary and hinge, whilst haltere development requires widespread Hox (Antp or Ubx) expression? This then is more like patterning of tissue/organ identity by absence of Hox expression, somewhat contrary to the usual consensus that presence of specific Hox codes provide identity.

3) Are then the phenotypic changes observed in the wings and halteres really due only to changes in Hox levels, as the authors have currently implied? Or is it more accurate to say that the phenotypic changes are due to changing the spatial expression profiles of the proteins, for example expanding Antp from the wing boundary to dorsal and ventral compartments of the wing pouch to make 'haltere'? That is, what is more important for the specification of wing versus haltere, the levels of Hox (Antp) expression in the wild type domain, or the spatial restriction of the expression to the wild type domain?

Whilst I can see (e.g. from Fig.3) that varying the levels of Hox makes the respective phenotypic changes more pronounced, this effect is not really distinguishable from the correlated changes to the spatial expression patterns.

Also, in Fig.3H is it fair to say that as the Ubx levels are reduced to the extent produced by *MS>UbxRNAi* at 25°C, there is still some weak Ubx expression in a line across the pouch that might be comparable/analogous to the wing boundary expression of Antp in T2?

In summary, I am wondering if it is more accurate to interpret these results as indicating that the difference between wing and haltere development is due to a combination of spatially heterogeneous expression of Hox (either Antp or Ubx) proteins across the respective appendage

disc (wing or haltere), in combination with a levels effect, such that at low Hox levels there is a wing-like expression (weak Hox across the boundary and undetectable in the dorsal and ventral pouch compartments) and at high levels there is a haltere-like expression (with Hox now also detectable in the dorsal and ventral pouch areas). Thus, the authors are not wrong to say that levels are important. However, they perhaps also need to acknowledge that in this wing-haltere system there might also be a functionally relevant spatial effect as well, which is intimately linked with varying the expression levels.

Abstract, first sentence: it is not entirely true to say flying insects have invaded all niches. There are no marine flying insects!

There are also some minor typos for correction, as follows.

- 4) Line 58: correct spelling of 'mellifera'. And check this spelling mistake throughout the manuscript.
- 5) Lines 66 and 67 and throughout: please use the accepted scientific convention of decimal points ('.') instead of commas.
- 6) Line 69: change 'are challenging' to 'challenge'.
- 7) Line 73: change 'Pioneer' to 'Pioneering'.
- 8) Line 82: insert 'a' to read, "in a few".
- 9) Line 351: remove the 'd' to give "harmonize".
- 10) Expanded view 9: for the MS1096>HthRNAi row in panel (A), it looks as though the order of the enlarged views has been switched and the Hth image has been put in the middle rather than on the top.

David Ferrier

Reviewer #2:

Remarks to the Author:

Nature Communications manuscript NCOMMS-20-44230-T

Paul et al

The manuscript by Paul et al describes an analysis of the expression and functions of the Hox genes Ubx and Antp during wing appendage formation. They use *Drosophila* as a genetic model and carry out comparisons across three additional insect Orders. The authors first show that Antp is expressed in wing precursors in all organisms, a somewhat surprising result given that the general understanding is that there is no Hox gene expression in wing-forming T2 cells. They also show that levels of Hox gene expression vary amongst different species with different T2-T3 arrangements of wing appendages. The authors then go on to genetically confirm the requirement for Antp in wing formation in *Drosophila*, and in an interesting set of experiments provide evidence that it is the levels of Hox proteins (either Antp or Ubx) that determine wing versus haltere fate, with higher levels promoting haltere fate. The main important results here are the demonstration of Antp expression and function in wing development, and the implication that overall Hox levels (rather than which particular Hox gene is expressed) control wing appendage fate.

These findings should be taken into context with some concerns that I have. In particular, it is not clear how the data in Figure 1 relate to the rest of the paper. In Figure 1, the authors show

drastically varying levels of T2/T3 Hox gene expression between Odonata and Lepidoptera, however the relative sizes of fore- and hind-wings is similar. This does not appear to support the model generated from the remainder of the paper.

In addition, the data in Figure 2A show that Antp levels in the wing pouch vary considerably over developmental time period, and it is not clear (a) whether the genetic manipulations affect the early, mid- or late expression of Antp and Ubx; (b) which is the critical stage for determining wing appendage size; and (c) whether the stages used in Figure 1 to assess Hox gene expression in wing precursors is the critical time for specification of wing size or fate.

Figure 4 does not add a great deal to the paper, once again because there is not an obvious pattern to the levels of Hox gene expression and the type or size of wing appendage formed.

Additional comments:

1. It would be good to see data on the levels of Antp in the Antp CRISPR mutants in Figure 2D.
2. The paper is overall rather short and it was frustrating to have to refer so many times to supplementary work, so it would be good to have more of the supplementary data in the main manuscript. It would be good to see the FGF data in the main paper.

Reviewer #3:

Remarks to the Author:

This paper analyzes the role of two Hox genes, Antp and Ubx, in the control of flight appendage (wings, halteres) diversity among insects. They first show different levels of Antp and Ubx expression in the forewing and hindwing of various four-winged insect species to find a correlation between the overall gene dose given by these two Hox genes and the relative sizes of the insect wings. They then use *Drosophila* genetics to test their hypothesis of the role that different levels of Hox gene expression play in the evolution of the flight appendages.

I find the work well done, most particularly in what concerns the *Drosophila* experiments, which show that high levels of either Antp or Ubx hinders wing development, converting the appendage into a haltere. They also show that this effect is dose dependent. However, I find that this work fails to support their main conclusion, namely, the direct involvement of Hox gene dosage in the evolution of the flight appendages of insects. In particular, if the results from their *Drosophila* experiments were relevant to the development of flight appendages in the insects for which they had measured Hox expression levels in the first part of the manuscript, it would be expected that none of these animals would have wings, as in all of them both forewings and hindwings have very high Hox (Antp and Ubx) expression levels. Actually, the observation that the three species analyzed develop wings from their T3 segment despite high Ubx expression indicates that, either the Ubx protein in these species is functionally different from that in *Drosophila* or that the cell environment in T3 make this segment respond differently in *Drosophila* and in the other three insect species.

Point-by-point response to the reviewers (all reviewer's comments are in italic, with major concerns underlined):

Reviewer-1

Our interpretation: the reviewer evaluated our manuscript of widespread interest with experiments carefully done and clearly documented. The reviewer had concerns about our conclusions outside *Drosophila*, and more particularly when considering results obtained in the silkworm *Bombyx mori*.

*Hox dosage is a rather over-looked aspect to Hox function and so the current manuscript is providing an important insight into these major developmental control genes. The authors also provide new expression data for the Antp protein in wing development of *Drosophila melanogaster*, which is impressive given the number of people that have studied this gene over a good number of years. This new expression data helps to reconcile some previously observed effects of Antp mutants in the wing development, which is satisfying.*

I think that the results reported in this manuscript will be of widespread interest, to mechanistic developmental biologists, particularly those working on the Hox genes, as well as evolutionary developmental biologists, who will be pleased to see the inclusion of the inter-species comparisons made here. There are a wealth of new results reported in this manuscript and the experiments are carefully done and clearly documented.

My main comments are that the conclusions that can be drawn from these results might be a bit more nuanced than the authors currently describe.

*1) The 'story' is not as simple as expression levels, as shown by Fig.1. Specifically, it is the *Bombyx mori* data that complicates matters. The damsel-fly data fits with the hypothesis that similar levels of Hox (Antp+Ubx) lead to similar sized wings, whilst *Apis mellifera* shows higher Hox levels (when Antp is summed with Ubx) in the hindwing correlating with a smaller wing size than the fore-wing. *Bombyx mori*, however, has much higher Hox levels in the HW, but not much of a size reduction of the HW. I missed a discussion of this apparently contradictory *Bombyx* result.*

This interpretation on *Bombyx mori* data is due to a focus on the wing size, probably due to an unclear presentation from our side (by keeping a short format version with no discussion section). We tried to emphasize more clearly that wing morphology is not only size, but also shape, and that therefore the Hox dose could impact on one of this aspect and not obligatory the two, as observed in *B. mori*. We now clearly outlined in the revised version (by

modifying the last model figure, now Figure 8, and discussing this point in the discussion section) that a change in the Hox dose between FW and HW primordia could result in morphological differences at the level of the shape, and not obligatory the size: “In addition, the overall quantification of Antp and Ubx expression level in the two flight primordia showed a striking correlation with the presence of distinct or similar pairs of wings on the T2 and T3 segments. This correlation is evident at the level of wing size in the damselfly and bee, while the distinct expression level observed in *B. mori* correlates with different wing shapes. These various impacts could potentially be due to various temporal requirements, as observed in our temperature restrictive assays in *Drosophila*. For practical reasons, quantifications have been performed at late stages in the different species, and this aspect could therefore not properly be addressed. In addition, these stages are not equivalent and the expression patterns and levels cannot be compared between the different species (the Q-PCR conditions are not identical and the affinity of Antp and Ubx antibodies is not known in the different species).”.

*2) The ability of Antp to specify a haltere in T3 and to rescue haltere phenotype in the *abx-pbx-bx* mutants seems surprising, and perhaps counterintuitive (since Ubx is supposed to be the ‘haltere-specifying’ Hox gene). Does this mean that wing development requires very restricted Hox (Antp) expression, such as just along the wing boundary and hinge, whilst haltere development requires widespread Hox (Antp or Ubx) expression? This then is more like patterning of tissue/organ identity by absence of Hox expression, somewhat contrary to the usual consensus that presence of specific Hox codes provide identity.*

This point (and others) raised the potential importance of the distinct Antp and Ubx expression profiles in the L3 wing and haltere discs, respectively. This is clearly not neutral, and our temperature-restrictive assays now show a correlation between the expression profile of Antp along the dorsal/ventral boundary of the wing disc at the L1 and L2 stages and its requirement for correct wing margin formation during these larval stages (new Fig. 2). In addition, the more ubiquitous expression at the L1 stage correlates with a more pronounced role for growth of the wing disc, a role that is required across all larval stages (and suggesting that the expression of Antp in the dorsal and ventral sides of the wing disc pouch at the L3 stage is mostly required for growth).

3) Are then the phenotypic changes observed in the wings and halteres really due only to changes in Hox levels, as the authors have currently implied? Or is it more accurate to say that the phenotypic changes are due to changing the spatial expression profiles of the

proteins, for example expanding Antp from the wing boundary to dorsal and ventral compartments of the wing pouch to make 'halterere'? That is, what is more important for the specification of wing versus haltere, the levels of Hox (Antp) expression in the wild type domain, or the spatial restriction of the expression to the wild type domain?

Whilst I can see (e.g. from Fig.3) that varying the levels of Hox makes the respective phenotypic changes more pronounced, this effect is not really distinguishable from the correlated changes to the spatial expression patterns.

Also, in Fig.3H is it fair to say that as the Ubx levels are reduced to the extent produced by MS>UbxRNAi at 25°C, there is still some weak Ubx expression in a line across the pouch that might be comparable/analogous to the wing boundary expression of Antp in T2?

In summary, I am wondering if it is more accurate to interpret these results as indicating that the difference between wing and haltere development is due to a combination of spatially heterogenous expression of Hox (either Antp or Ubx) proteins across the respective appendage disc (wing or haltere), in combination with a levels effect, such that at low Hox levels there is a wing-like expression (weak Hox across the boundary and undetectable in the dorsal and ventral pouch compartments) and at high levels there is a haltere-like expression (with Hox now also detectable in the dorsal and ventral pouch areas). Thus, the authors are not wrong to say that levels are important. However, they perhaps also need to acknowledge that in this wing-haltere system there might also be a functionally relevant spatial effect as well, which is intimately linked with varying the expression levels.

This point also relates to the potential importance of the expression profile, which we originally underlined but probably not sufficiently in the description of the wing-like Antp expression pattern in the *Ubx* null background of the transformed haltere (Figure 5 in the revised version). There is clearly a non-neutral role of intra-pouch specific expression pattern for distinguishing Antp and Ubx, in addition to the dose. This is now more clearly stated in the discussion section: "Finally, the observation of a specific Antp expression profile in the T2 wing disc pouch, which is also found in the haltere-to-wing transformed tissue disc of *Ubx* mutant individuals, strongly suggests that the distinct roles of Antp and Ubx in the wing and haltere discs are also linked to a distinct expression profile in addition to specific doses". Although focusing on the dose, the role of the expression profile is now also mentioned in the legend of the Figure 8.

Finally, to assess for the role of the dose only, we tried several times to use the *P1-Antp-Gal4* driver to overexpress Antp while respecting its expression profile in the wing disc.

Unfortunately, this combination was lethal, even when combined with the *Tub-Gal80^{ts}*

chromosome to drive the expression at specific larval stages (from L1 to L3), suggesting that this driver is also expressed in other tissues that are sensitive to the level of Antp for proper development (likely the nervous system). This negative result was not mentioned in the revised manuscript.

Abstract, first sentence: it is not entirely true to say flying insects have invaded all niches. There are no marine flying insects!

True! We changed the first sentence of the abstract accordingly.

There are also some minor typos for correction, as follows.

4) Line 58: correct spelling of 'mellifera'. And check this spelling mistake throughout the manuscript.

5) Lines 66 and 67 and throughout: please use the accepted scientific convention of decimal points ('.') instead of commas.

6) Line 69: change 'are challenging' to 'challenge'.

7) Line 73: change 'Pioneer' to 'Pioneering'.

8) Line 82: insert 'a' to read, "in a few".

9) Line 351: remove the 'd' to give "harmonize".

10) Expanded view 9: for the MS1096>HthRNAi row in panel (A), it looks as though the order of the enlarged views has been switched and the Hth image has been put in the middle rather than on the top.

All minor typos have been corrected.

Reviewer-2

Our interpretation: this reviewer had the same general concern about the correlation that we tried to establish between results obtained in *Drosophila* and observations in the other insect species.

The manuscript by Paul et al describes an analysis of the expression and functions of the Hox genes Ubx and Antp during wing appendage formation. They use Drosophila as a genetic model and carry out comparisons across three additional insect Orders. The authors first show that Antp is expressed in wing precursors in all organisms, a somewhat surprising

result given that the general understanding is that there is no Hox gene expression in wing-forming T2 cells. They also show that levels of Hox gene expression vary amongst different species with different T2-T3 arrangements of wing appendages. The authors then go on to genetically confirm the requirement for Antp in wing formation in *Drosophila*, and in an interesting set of experiments provide evidence that it is the levels of Hox proteins (either Antp or Ubx) that determine wing versus haltere fate, with higher levels promoting haltere fate. The main important results here are the demonstration of Antp expression and function in wing development, and the implication that overall Hox levels (rather than which particular Hox gene is expressed) control wing appendage fate.

These findings should be taken into context with some concerns that I have. In particular, it is not clear how the data in Figure 1 relate to the rest of the paper. In Figure 1, the authors show drastically varying levels of T2/T3 Hox gene expression between Odonata and Lepidoptera, however the relative sizes of fore- and hind-wings is similar. This does not appear to support the model generated from the remainder of the paper.

As previously mentioned with the Reviewer 1, we now more clearly stated in the revised version that dose variation could differently impact on the wing morphology, from size and shape (like in *Apis* or *Drosophila*) to shape only (like in *B. mori*). We would like to further strengthen here that the morphology (size and/or shape) is systematically different when correlated to a different Hox dose between the FW and HW primordia. Along this line, local-high Antp expression has been described to be associated with the formation of pigmentation spots in the FW and HW of butterflies (*Saenko et al., EvoDevo 2011*), highlighting that Hox dose could also have a role on other subtle wing morphological traits in insects.

- In addition, the data in Figure 2A show that Antp levels in the wing pouch vary considerably over developmental time period, and it is not clear (a) whether the genetic manipulations affect the early, mid- or late expression of Antp and Ubx; (b) which is the critical stage for determining wing appendage size; and (c) whether the stages used in Figure 1 to assess Hox gene expression in wing precursors is the critical time for specification of wing size or fate.

Points (a): the time window of *Nub-Gal4* and *MS-1096* drivers used to manipulate the dose of Antp or Ubx is mentioned in the main text or the legend of the Fig.2, respectively. *MS-1096* starts at the L1 stage while *Nub-Gal4* starts at 2nd instar stage. Genetic manipulations with these drivers therefore affect early and late Hox expression in the wing and haltere imaginal discs.

Point (b): we included a time-restrictive assay in the new Figure 2 (Fig. 2D') to show that Antp is required from L1 to L3 stages (with a contribution for both wing size and wing margin at L1 and L2 stages, and a contribution for wing size only at the L3 stage).

Point (c): we mentioned the stages of wing primordia dissection for each species in the legend of the Figure 7, which are late stages, due to practical considerations (wing primordia were too small and fragile at earlier stages, and a certain quantity was necessary for the Q-PCRs). We discussed this point in the discussion section, saying that "For practical reasons, quantifications have been performed at late stages in the different species, and this aspect could therefore not properly be addressed. In addition, these stages are not equivalent and the expression patterns and levels cannot be compared between the different species (the Q-PCR conditions are not identical and the affinity of Antp and Ubx antibodies is not known in the different species)". We are also more precautionous in the way we present the model: "In any case, although our observations in four-winged species need further genetic validation, we propose a speculative model where variation in the Hox dosage (and possibly the profile) could be used to diversify flight appendage morphology, from subtle wing size and/or shape modification to the formation of a completely new organ during insect evolution (Fig. 8)".

- Figure 4 does not add a great deal to the paper, once again because there is not an obvious pattern to the levels of Hox gene expression and the type or size of wing appendage formed.

We emphasized that wing morphology is not only size and we hope that the reviewer will be convinced that Hox dosage could be involved in more subtle phenotypic traits such as wing shape. We modified the model (new Fig. 8) to better highlight the correlation between Hox dose and the different types of morphological traits in insects. Although speculative, this model fits to our observations and we believe that it constitutes an interesting and stimulating opening for future work.

- Additional comments:

1. It would be good to see data on the levels of Antp in the Antp CRISPR mutants in Figure 2D.

We now show the loss of Antp expression when using our Crispr/Cas9-mediated tools in the wing disc pouch in the new Fig. 2 (Fig. 2A' and 2B').

2. *The paper is overall rather short and it was frustrating to have to refer so many times to supplementary work, so it would be good to have more of the supplementary data in the main manuscript. It would be good to see the FGF data in the main paper.*

The first version was indeed *Nature*-formatted. We completely reformatted the manuscript, with many more figures and less supplementary figures. The expression of wing-specifying genes is now shown in the new Fig.3. As mentioned in the Editor letter, because reviewers all agree on the importance of results in *Drosophila*, we started by this part in the revised version, and finished on the analysis in the other species with the speculative model at the very end.

Reviewer 3

Our interpretation: this reviewer also found the work well done, especially the part related to the experiments in *Drosophila*. He/she had concerns about our conclusions raised from observations in the other insect species, but for different reasons than the ones mentioned by the two other reviewers: the reviewer underlined that the Hox expression level was quite high in the other four-winged species, which was somehow counter-intuitive when considering the importance of the low Hox dose for wing formation in *Drosophila*.

This paper analyzes the role of two Hox genes, Antp and Ubx, in the control of flight appendage (wings, halteres) diversity among insects. They first show different levels of Antp and Ubx expression in the forewing and hindwing of various four-winged insect species to find a correlation between the overall gene dose given by these two Hox genes and the relative sizes of the insect wings. They then use Drosophila genetics to test their hypothesis of the role that different levels of Hox gene expression play in the evolution of the flight appendages.

I find the work well done, most particularly in what concerns the Drosophila experiments, which show that high levels of either Antp or Ubx hinders wing development, converting the appendage into a haltere. They also show that this effect is dose dependent. However, I find that this work fails to support their main conclusion, namely, the direct involvement of Hox gene dosage in the evolution of the flight appendages of insects. In particular, if the results from their Drosophila experiments were relevant to the development of flight appendages in the insects for which they had measured Hox expression levels in the first part of the manuscript, it would be expected that none of these animals would have wings, as in all of them both forewings and hindwings have very high Hox (Antp and Ubx) expression levels.

Actually, the observation that the three species analyzed develop wings from their T3 segment despite high Ubx expression indicates that, either the Ubx protein in these species is functionally different from that in *Drosophila* or that the cell environment in T3 make this segment respond differently in *Drosophila* and in the other three insect species.

We agree that this point could be confusing, and we now carefully mentioned this aspect in the discussion section of the revised version: “For practical reasons, quantifications have been performed at late stages in the different species, and this aspect could therefore not properly be addressed. In addition, these stages are not equivalent and the expression patterns and levels cannot be compared between the different species (the Q-PCR conditions are not identical and the affinity of Antp and Ubx antibodies is not known in the different species). Still, the observation that Antp could be expressed in addition to Ubx in the HW primordium of four-winged species and not in *Drosophila* suggests that wing developmental programs could be differently sensitive to the Hox dose in different insect species”.

Reviewers' Comments:

Reviewer #1:

Remarks to the Author:

The authors have done a good job in addressing my previous comments. My further comments are all minor in nature.

In reference to my original point (3) – the authors tried to do an experiment that only changed the dosage of Antp and not the spatial profile, but this proved technically impossible. The authors have revised their discussion to reflect this. That is, dosage and spatial profiles are intimately linked and both may be operating in concert to determine and pattern the flight appendages.

1) The authors seem to be using the word 'profile' to represent a spatially heterogenous pattern. I am not sure that this is the best choice of word, at least when it is used on its own, because profile can include all sorts of aspects of expression (including time and levels). If the aim is to draw a distinction between dosage/levels and spatial expression then this should be specifically articulated.

A specific couple of instances of this are on lines 206-7, where 'distinct expression profile' should be changed to 'distinct spatial expression profiles'. Also, a similar change is required on line 229.

2) Title – since there is still some ambiguity about whether Hox dosage is the sole explanation for causing distinctive flight appendage morphogenesis, instead of a combination of dosage with other factors such as particular spatial profiles, then perhaps the title needs a minor adjustment to reflect this. How about, "A role for Hox dosage in the control of flight appendage morphology in Drosophila"? Or "Hox dosage contributes to flight appendage morphology in Drosophila" – the point here is that the word 'controls' implies that dosage is the primary, and maybe even the only, factor controlling flight appendage development, but this is not demonstrated in this manuscript.

3) Line 37: can delete "in the air" as this is implicit in the word 'flying'.

4) Line 133: change 'reversely' to 'conversely'.

5) Line 158: change 'resembled to a' to 'resembled a'.

6) Line 164: change 'intermediary' to 'intermediate' in both cases.

7) Line 169: change 'fore-wings' to 'fore-winged'.

8) Fig6. It appears that the UbxRNAi does not simply knock down Ubx uniformly – or at least the expression does not go from uniformly high to uniformly low. Instead, the knock-down appears to produce an expression pattern with some Ubx still present along the margin(?) region of the disc/pouch with lower levels in the dorsal and ventral domains. Thus, not only has Hox dosage been reduced, but the spatial pattern has a more wing-like appearance instead of a haltere-like appearance. The dosage and spatial profiles seem to be intimately linked. If the authors agree, this should be clearly described, in a similar way to how the authors have revised their emphasis from only being on levels to now also incorporating other aspects of expression (such as distinctive spatial profiles) elsewhere.

9) Fig 8 says that the fore-wings of damselflies, moths and bees expresses both Antp and Ubx, but the Fig7 data looks to indicate that Ubx is not expressed in fore-wings in these three species.

Reviewer #2:

Remarks to the Author:

The authors have made significant changes to the organization and content of their manuscript, which has very effectively improved its organization and logical flow, and more readily describes its impact. The focus upon genetic manipulations in *Drosophila* emphasizes the mechanistic findings, and the analyses of Hox gene expression in other insects provide potential generalization of the results to other species. Other than indicating the dorsal and ventral wing blade compartments on Figure 1, I have no additional changes to suggest.

Reviewer #3:

Remarks to the Author:

In this revised manuscript the authors have changed the focus of their work, now centering it in the analysis of how alteration of Hox levels affect wing development from the thoracic segments of *Drosophila*. The possible implications of the *Drosophila* work on the evolution of wing patterns in insects are now carefully commented, making sure that their genetic experiments in *Drosophila* do not come into conflict with Hox expression data in four-winged species. My objections to the previous version of this work have been fully addressed and I think that the work is now solid and well balanced.

Point-by-point response to the reviewers' comments.

Reviewer #1 (Remarks to the Author):

The authors have done a good job in addressing my previous comments. My further comments are all minor in nature.

In reference to my original point (3) – the authors tried to do an experiment that only changed the dosage of Antp and not the spatial profile, but this proved technically impossible. The authors have revised their discussion to reflect this. That is, dosage and spatial profiles are intimately linked and both may be operating in concert to determine and pattern the flight appendages.

1) The authors seem to be using the word 'profile' to represent a spatially heterogenous pattern. I am not sure that this is the best choice of word, at least when it is used on its own, because profile can include all sorts of aspects of expression (including time and levels). If the aim is to draw a distinction between dosage/levels and spatial expression then this should be specifically articulated.

A specific couple of instances of this are on lines 206-7, where 'distinct expression profile' should be changed to 'distinct spatial expression profiles'. Also, a similar change is required on line 229.

We agree with the reviewer that 'expression profile' was not clear enough and we systematically changed it into "spatial expression profiles" in the appropriate places (lines 217-218; 240 and 450 of the new submitted ms).

2) Title – since there is still some ambiguity about whether Hox dosage is the sole explanation for causing distinctive flight appendage morphogenesis, instead of a combination of dosage with other factors such as particular spatial profiles, then perhaps the title needs a minor adjustment to reflect this. How about, "A role for Hox dosage in the control of flight appendage morphology in Drosophila"? Or "Hox dosage contributes to flight appendage morphology in Drosophila" – the point here is that the word 'controls' implies that dosage is the primary, and maybe even the only, factor controlling flight appendage development, but this is not demonstrated in this manuscript.

The title has now been changed from ‘Hox dosage controls flight appendage morphology in *Drosophila*’ into ‘Hox dosage contributes to flight appendage morphology in *Drosophila*’, as suggested by the reviewer.

3) Line 37: can delete “in the air” as this is implicit in the word ‘flying’. Done.

4) Line 133: change ‘reversely’ to ‘conversely’. Done.

5) Line 158: change ‘resembled to a’ to ‘resembled a’. Done.

6) Line 164: change ‘intermediary’ to ‘intermediate’ in both cases. Done.

7) Line 169: change ‘fore-wings’ to ‘fore-winged’. Done.

8) Fig6. *It appears that the UbxRNAi does not simply knock down Ubx uniformly – or at least the expression does not go from uniformly high to uniformly low. Instead, the knock-down appears to produce an expression pattern with some Ubx still present along the margin(?) region of the disc/pouch with lower levels in the dorsal and ventral domains. Thus, not only has Hox dosage been reduced, but the spatial pattern has a more wing-like appearance instead of a haltere-like appearance. The dosage and spatial profiles seem to be intimately linked. If the authors agree, this should be clearly described, in a similar way to how the authors have revised their emphasis from only being on levels to now also incorporating other aspects of expression (such as distinctive spatial profiles) elsewhere.*

This is an interesting observation, which is explained by the expression profile of the *MS1096* driver (it is not strongly expressed along the wing margin). Although speculative, it is worth mentioning this aspect to further illustrate the potential role of the spatial Hox expression profile in addition to the dose. This is mentioned in the discussion section (lines 214-216): ‘*Interestingly, RNAi experiments with MS1096 led to residual Ubx expression along the DV boundary, recalling a spatial pattern that resembled to Antp in the wing disc.*’

9) Fig 8 says that the fore-wings of damselflies, moths and bees expresses both *Antp* and *Ubx*, but the Fig7 data looks to indicate that *Ubx* is not expressed in fore-wings in these three species.

There is residual Ubx level (from RT-qPCR) but we agree that it is finally negligible when compared to Antp level. We simplified the Fig.8, showing only Antp in the FW primordia, to better illustrate RT-qPCR data. The legend of the Fig.8 has been changed accordingly: *'The model is based on the Hox (Antp or Antp+Ubx) expression level in the forewing (FW) and hindwing (HW) primordia. Ubx level was systematically considered as negligible in the FW primordium of all studied species.'*

Reviewer #2 (Remarks to the Author):

The authors have made significant changes to the organization and content of their manuscript, which has very effectively improved its organization and logical flow, and more readily describes its impact. The focus upon genetic manipulations in Drosophila emphasizes the mechanistic findings, and the analyses of Hox gene expression in other insects provide potential generalization of the results to other species. Other than indicating the dorsal and ventral wing blade compartments on Figure 1, I have no additional changes to suggest.

The dorsal and ventral wing blade compartments are now indicated in the Fig.1a and 1b (with the corresponding change in the figure legend).

Reviewer #3 (Remarks to the Author):

In this revised manuscript the authors have changed the focus of their work, now centering it in the analysis of how alteration of Hox levels affect wing development from the thoracic segments of Drosophila. The possible implications of the Drosophila work on the evolution of wing patterns in insects are now carefully commented, making sure that their genetic experiments in Drosophila do not come into conflict with Hox expression data in four-winged species. My objections to the previous version of this work have been fully addressed and I think that the work is now solid and well balanced.

This reviewer was fully satisfied by the submitted version and had no additional comment.